# Genomics of Wolfram Syndrome 1 (WFS1)

**DOI:** 10.3390/biom13091346

**Published:** 2023-09-04

**Authors:** Sulev Kõks

**Affiliations:** 1Perron Institute for Neurological and Translational Science, 8 Verdun Street, Nedlands, WA 6009, Australia; sulev.koks@uwa.edu.au; 2Centre for Molecular Medicine and Innovative Therapeutics, Murdoch University, Perth, WA 6150, Australia

**Keywords:** Wolfram syndrome, *WFS1* gene, wolframin protein, genomics, functional genomics

## Abstract

Wolfram Syndrome (WFS) is a rare, autosomal, recessive neurogenetic disorder that affects many organ systems. It is characterised by diabetes insipidus, diabetes mellites, optic atrophy, and deafness and, therefore, is also known as DIDMOAD. Nearly 15,000–30,000 people are affected by WFS worldwide, and, on average, patients suffering from WFS die at 30 years of age, usually from central respiratory failure caused by massive brain atrophy. The more prevalent of the two kinds of WFS is WFS1, which is a monogenic disease and caused by the loss of the *WFS1* gene, whereas WFS2, which is more uncommon, is caused by mutations in the *CISD2* gene. Currently, there is no treatment for WFS1 to increase the life expectancy of patients, and the treatments available do not significantly improve their quality of life. Understanding the genetics and the molecular mechanisms of WFS1 is essential to finding a cure. The inability of conventional medications to treat WFS1 points to the need for innovative strategies that must address the fundamental cause: the deletion of the *WFS1* gene that leads to the profound ER stress and disturbances in proteostasis. An important approach here is to understand the mechanism of the cell degeneration after the deletion of the *WFS1* gene and to describe the differences in these mechanisms for the different tissues. The studies so far have indicated that remarkable clinical heterogeneity is caused by the variable vulnerability caused by *WFS1* mutations, and these differences cannot be attributed solely to the positions of mutations in the *WFS1* gene. The present review gives a broader overview of the results from genomic studies on the WFS1 mouse model.

## 1. Introduction

**Wolfram Syndrome** (WFS) is a rare, autosomal, recessive neurogenetic disorder that affects different organs and functions. The syndrome is characterised by diabetes insipidus, diabetes mellites, optic atrophy, and deafness, hence the acronym **DIDMOAD** [1,2,3]. It is inherited as an autosomal recessive disease caused by homozygous or compound heterozygous mutation in the *WFS1* gene that encodes the wolframin ER transmembrane glycoprotein [4,5,6]. Wolfram syndrome 2 is caused by the homozygous mutations in the *CISD2* gene and is also inherited as an autosomal recessive disorder [7]. The *WFS1* gene is located in chromosome 4, locus 4p16.1 and *CISD2* is in 4q24 [4,5,7,8]. WFS can be diagnosed as juvenile diabetes mellitus and diabetes insipidus, combined with deteriorating vision and hearing in a child with no developmental abnormalities and perfectly normal health until the onset [3,9,10]. Additional abnormalities start to develop, like urinary tract and bladder atonia, renal functional changes, and cataracts. The disease also affects the brain and central nervous system, and many neurological functions are affected, indicating a string impact in the neurodegeneration [3,11,12]. The syndrome can be very variable, but the presence of the DIDMOAD complex is central for the diagnosis of WFS1 [3]. In addition, the central neurodegenerative changes, neuromuscular pathology with biochemical alterations in muscles, have been described [13,14]. The complexity of clinical traits indicates that while the inheritance can be quite simple and clear, the penetrance and expression of the features is complex. On the other hand, we need to consider the complexity of the function of these proteins, WFS1 and CISD2. Lack of the function of reduced activity of the protein will impact specific functions in different cell types, leading to the complex clinical phenotype. Therefore, single mutation on a single gene can lead to a complex and diverse clinical presentation, depending on the affected cells, their resilience, and their functional capacity. Describing and understanding the mechanisms of this complexity forms the principal foundation of genomic medicine and genomic pathology. In the present review, we focus only on the *WFS1* gene and will give an overview of the recent genomic data for the *WFS1* gene.

## 2. *WFS1* Gene

WFS1 is caused by homozygous or compound heterozygous mutations in the *WFS1* gene [4,5,6,15]. Heterozygous carriers do not have any clinical features, but they have been repeatedly reported to increase susceptibility to variable clinical traits that are parts of the full spectrum of WFS1 [16,17,18,19]. It is well reported that heterozygous mutations in the WFS1 gene cause deafness or increase the risk for type 2 diabetes [20,21,22,23]. WFS1 heterozygosity is a well-established genetic risk factor for the metabolic disorders, psychiatric disorders, and even suicide [24,25,26,27,28]. The mechanism of how the *WFS1* gene causes neuropsychiatric manifestations is not clear, but the ER stress seems to be central and leads to the deficiency in the regulation of emotions [29]. This helps to explain the variable neuropsychiatric symptoms of WFS1 patients and the increased risk for emotional disorders in the carriers of *WFS1* mutations. The *WFS1* gene and its involvement in ER stress and cellular survival positions the gene to the centre of the brain and the endocrine pancreas axis [29]. The involvement of this gene in the ER stress can help to explain the co-morbidity between mood disorders and diabetes [30]. Interestingly, the WFS1 gene is highly expressed in the pancreas and brain, giving more support to its common function between the brain and pancreas bridging the functionally of these two organs [31]. Moreover, WFS1 is also highly expressed in the heart and skeletal muscles, the tissues that are also affected in WFS1-mutant animals, but not reported so often in humans [14,32]. Cloning of the *WFS1* gene gave us the understanding of the structure of the gene and helped to pinpoint the locations of the mutations [4,33]. The *WFS1* gene codes an 890-amino acid transmembrane protein with the molecular mass 100 kD. WFS1 protein has nine transmembrane domains and localises in the endoplasmic reticulum (ER) membrane. The C-terminus of the WFS1 is positioned in the ER lumen, and the N-terminus is cytoplasmic [4,34,35]. It appears the WFS1 protein is assembled into higher molecular complexes, and it forms tetramers with the molecular mass 400 kD [35].

The *WFS1* gene has eight exons (Figure 1), with the eighth exon as the largest, and the gene can form 15 different transcripts (Table 1). Most of the mutations in the *WFS1* gene are at exon 8 [33,36]. As shown in Figure 2, while the genomic variants are spread all over the *WFS1* gene, the variants with pathogenic, likely pathogenic, and uncertain significance are mostly accumulated in exon 8. Only a few pathogenic/likely pathogenic variants are in the other parts of the gene. Interestingly, for the full-spectrum Wolfram syndrome, both alleles must be affected by the mutations. In the case of only single-allele mutation, significantly increased risk for type 2 diabetes, hearing loss, depression, suicide, and other complex diseases have been described. While the list of these diseases is long, the phenotypic complexity of WFS1 helps to explain most of these traits affected by the variations in the *WFS1* gene. The most described findings with the heterozygosity of the *WFS1* gene are related to the hearing loss and psychiatric diseases. One of the most remarkable effects is related to the psychiatric illnesses. It has been identified that out of 11 relatives of WFS1 patients who were hospitalised, 10 were heterozygous carriers of *WFS1* mutations [17]. This indicated a 26-fold increased risk for the psychiatric hospitalisation of WFS1 carriers versus non-carriers. Psychiatric findings are very common in WFS1 patients and relatives, indicating the significance of the *WFS1* gene as a potential target for psychiatric illnesses. On the other hand, the effect of WFS1 mutations on growth has not been described as a common feature of the disease. This is peculiar, as the pituitary atrophy in WFS1 patients was described a long time ago [37]. Moreover, the short stature and growth hormone deficiency has also been reported in early studies on WFS1 patients [38]. Mouse studies also identified the growth retardation after the deletion of the *WFS1* gene [39].

## 3. WFS1 Mutations and Pathophysiology

The function of the *WFS1* gene is still not fully understood. The protein is involved in ER stress, protein misfolding control, and proteostasis, but the exact mechanism is not known [8]. Recent studies suggest that WFS1 participates in mitochondria-associated ER membranes (MAMs), co-localises, and forms a complex with neuronal calcium sensor 1 (NCS1) [40,41]. This way, the WFS1 protein participates directly in the mitochondria and ER crosstalk that is essential for the cell metabolism and survival. This large impact on cell physiology explains why WFS is a complex syndrome that affects many different systems.

It is important to stress here that WFS1 mutations do not impair the mitochondria and mitochondrial morphology directly [41]. WFS1 protein seems to be involved in the mitochondrial Ca^2+^ uptake by an inositol 1,4,5-trisphosphate receptor (IP_3_R) pathway, and the reduced response to IP_3_ has been shown [41]. However, the molecular action of WFS1 on mitochondria is not that direct and seems to be mediated by the interactions among IP_3_R, voltage dependent anion channel 1 (VDAC1), and glucose-regulated protein 75 (GRP75), also known as heat shock protein family A (Hsp70) member 9 (HSPA9) [41]. These findings suggest a more widespread impact on the vesicular traffic and ER–mitochondrial connection. Moreover, WFS1’s interaction with neuronal calcium sensor 1 (NCS1) was confirmed and proposed to be fundamental for this complex interaction of WFS1 and its functional partners. NCS1 interaction with WFS1 can also explain the respiratory chain changes seen in WFS1 patients and mutant mouse muscles [14,41]. In addition, knockdown of the *WFS1* gene caused changes in TOMM20, another protein associated with ER–mitochondrial transport [42]. Therefore, the mutations in the *WFS1* gene do not impact mitochondria directly. The influence on mitochondrial function is realised through a complex interaction with different protein complexes, and ER–mitochondrial crosstalk is fundamental for the pathophysiology of the syndrome.

WFS1 mutations usually have a drastic effect on the clinical phenotype, and to have a full WFS1 homozygosity or compound heterozygosity is necessary. It has been shown that the mutations not only impact the WFS1 protein, but many effects emerge at the RNA level. For instance, patients with the frameshift mutation or with the compound heterozygous stop and missense mutation both had clinically proven WFS1 [35,43]. The frameshift mutations would have given rise to the truncated proteins, but in the patient with these mutations, no truncated protein was found. This shows that frameshifts in the *WFS1* gene lead to the complete absence of the protein, explaining the high penetrance of these mutations. And, even in the patient with compound heterozygosity with a stop and missense variant, only 5% of the WFS1 protein was detected [35,43]. This indicates that while the inactivation of the WFS1 protein is necessary for Wolfram syndrome, the mechanism of the inactivation is not clear.

The function of *WFS1* is related to neuropsychiatric disorders, and its expression profile is specific for the neuroendocrine pathologies [29]. Since the first study indicating that heterozygous WFS1 mutation carriers have increased risk for mood disorders, many additional confirmative studies have been published [16,17,44]. Moreover, variants in the *WFS1* gene have been shown to be involved in the higher risk of suicide [24,26] These studies also suggest the widespread neurodegeneration in WFS patients, and these results can explain the role of the stress system in the depression and neurodegeneration [45]. It has been postulated that the missense variants cause WFS1 protein to form insoluble aggregates as a primary mechanism for the degeneration [46]. However, the *WFS1* RNA expression is also significantly reduced in the case of the missense mutations, indicating more complex pathogenic regulation [43]. Missense mutations clearly reduce the stability of the protein and have a significantly shorter half-life compared to the wild-type WFS1 protein. It is obvious that the quantity of the WFS1 protein is also important in the pathogenesis of the disease, and reduced amounts of the protein cause the disease, similarly to the complete lack of WFS1. These initial findings were further confirmed by additional studies where the stability of the *WFS1* transcript of the missense and truncating mutations was shown with a significantly reduced amount of WFS1 protein [43]. Therefore, while the stability of the transcripts seems not to be affected by mutations, mutated proteins undergo premature proteasomal degradation.

## 4. Wolfram Syndrome Animal Model

The main challenge with the rare disease is the lack of sufficient biomaterial for the biochemical and molecular studies, and this is the situation in which animal models become very useful. We developed the *WFS1* mutant mouse line many years ago, and it was based on conventional gene targeting, where most of the coding sequence was replaced by the LacZ cassette [47]. This mouse line very accurately recapitulated all aspects of WFS1, starting with the diabetes, infertility, eye problems, etc. [39,47,48,49,50,51,52,53,54]. Other rodent models for WFS1 syndrome have also been developed, but none of them recapitulates all the facets of WFS1 [55,56]. It is interesting that partial deletion of the *WFS1* gene in animals results only in partial syndromes, and often these syndromes have very weak expression of traits and lack important parts of WFS1. The establishment of the mouse WFS1 model allowed us to perform series of genomic studies and to understand more about the complexity of the genomic regulation after deletion of the *WFS1* gene.

The mouse model for the WFS1 syndrome that lacked two-thirds of the *WFS1* gene reproduced all clinical features of this complex disease, like diabetes and mental disorders [47]. Within many years, several studies based on this mouse model have been published to describe the variable phenotypic effects of the mutation [57,58]. Very briefly, we have identified the growth retardation and development of combined endocrine disorders, combined with diabetes, in *WFS1* mutant mice [39,48,50,59]. The growth retardation is the first trait of the mutation, and it starts to appear around 9 weeks of age [39]. This is one of the most cardinal features and is followed by slow development of all other traits characteristic of WFS1 syndrome in humans. Weight loss, diabetes, visual impairment, muscular dystrophy, neurochemical changes, and behavioural alteration follow at the later stages of life. Moreover, the lifespan of *WFS1* mutant mice is also shorter [60]. The knockdown of the *WFS1* gene in HEK cells induced the activation of mitochondrial damage and neurodegenerative pathways, suggesting the role of *WFS1* in the neurodegeneration [42]. These data indicate that WFS1 protein is involved in the neurodegeneration, and Wolfram syndrome is a systemic neurodegenerative disorder with complex symptoms.

## 5. Genomic Studies

Generation of the knockout or mutant mice is a complex experimental task that requires very good design to be successful at the end. This involves the design of the targeting construct, breeding the colonies, and design of the experiments to identify the traits that are affected by the mutation. One of the challenges is to avoid the genomic footprint effect, which can generate false positive phenotypes purely because of the random variations in the genomic background. We have addressed this question in several papers and have provided a solution for how to avoid randomly induced phenotypes that have nothing to do with the initial and planned mutation [39,61,62]. This approach is especially important when the phenotype is complex, like in the case of WFS1. Subtle traits can be missed if the experimental plan is not suitable or allows too much variability in the baseline phenotypes.

## 6. Transcriptomic Studies Using WFS1 Mouse Model

As we developed our own mouse model for WFS1, we had a great supply of the biomaterial for the variable genomic studies. This mouse line has been used by many other labs over the world, and this section attempts to summarise the main findings from the transcriptomic analyses [53,54,57,63]. Considering the expression pattern of the *WFS1* gene, the focus has been on the hypothalamus, hippocampus, eyes, and pancreatic islets [39,47,51,59]. The hypothalamus and hippocampus were analysed in two independent experiments using gene chips and RNA sequencing. We will compare these tissues separately and analyse the results of the two different methods for the transcriptomic analysis.

The analysis of hypothalamic samples with the gene chips did reveal an alteration in the G protein signalling pathway, with the changes in the regulator of the G protein signalling 4 (RGS4) and the regulator of the G protein signalling 16 (RGS16) genes [59]. RGS proteins have guanosine triphosphatase (GTPase)-activating protein (GAP) activity, and they are responsible for the rapid activation and deactivation of G-protein signalling, and the protein family contains more than 37 members [64]. But, most of the RGS proteins are not just GAPs, and they are structurally complex proteins: RGS protein includes the RGS7-like, RGS12-like, RhoGEF-containing, and G protein–coupled receptor kinase (GRK)-like classes [65]. They possess several protein–protein interaction domains that transduce signals to downstream effectors, and RGS proteins coordinate signalling between intracellular networks or mediate shuttling between intracellular compartments. RGS proteins have been recognised as crucial regulators and integrators for the seven-transmembrane domain receptor (7TMR) signalling [66]. In parallel to these findings in the WFS1 mutant mice, we have always detected alterations with other G-protein-related genes, like Rho GTPases, Golgi brefeldin A resistant guanine nucleotide exchange factor 1 (GBF1), or GTP-biding proteins. This certainly indicates the disturbances in the G-protein-related signalling and helps to explain the complexity of the phenotype. These changes were accompanied by the upregulation of the genes that are related to the proteasomal pathway and lysosomal degradation, indicating the mechanism of the cellular damage in WFS1. Interestingly, the RNA-sequencing base experiment did not find many overlaps in the gene names from the gene chip analysis, but this is quite expected, considering the differences in the dynamic ranges and sensitivity of the genomic analysis methods [67]. While we did not see much overlap in gene names, we did find the overlap of the genomic pathways activated between these two studies. Differential expression of multiple GTPases and several G protein-coupled receptors was detected (AVPR1A). Interestingly, FGFRL1, fibroblast growth factor receptor like 1, was found to be downregulated. FGFRL1 is a gene that regulates height and osteoporosis [68]. Pathway analysis identified that, in both studies, G protein signalling, the ER–stress pathway, and the proteosomal/lysosomal pathway were affected.

The analysis of the hippocampus was also performed in two independent studies, first with the gene chips and later by using RNA sequencing [39,51]. Again, while the list of differentially expressed genes overlapped only a little, the functional annotation of these two studies showed the activation of similar pathways. Both gene chip analysis and the RNA-sequencing-based experiment identified the expression of different guanine nucleotide-exchange factors (e.g., Golgi brefeldin, a resistant guanine nucleotide exchange factor 1 or GBF1) to be changed, indicating similar pathogenetic pathways [39,67]. GBF1 is a GTP exchange factor and plays a role in vesicular trafficking by activating ADP ribosylation factor 1 [69,70]. Moreover, GBF1 depletion induced ER stress, an unfolded protein response, and cell death in the cellular models, indicating a clear functional overlap with the *WFS1* gene [71]. We also detected different GTPases and several G protein-coupled receptors (GPR179, many olfactory receptors) to be differentially expressed. Like with previous studies, the proteasomal pathway and protein folding responsive genes (heat shock proteins) were also changed in these two independent studies. The growth hormone signalling appeared to be upregulated, as evidenced by the increased growth hormone transcription, and increased IGF-1 levels in the serum, despite the significantly smaller size of the *WFS1* mutant mice [39].

Interestingly, the hippocampal RNA sequencing analysis identified TRPM8 as upregulated in *WFS1* mutant mice. TRPM8, a menthol receptor, mediates thermal sensitivity and regulates metabolic activity, which led us to analyse the metabolic activity of *WFS1* mutant mice [28,60,67]. Indeed, we found that the metabolic activity of *WFS1* mutant mice was increased, and they were more responsive to the menthol [60].

The whole transcriptome analysis of the pancreatic islets from the WFS1 mice identified some genes that were identical to previously described studies, like neuronal PAS domain protein 4 and basic helix–loop–helix 15a protein. In the islets, we identified TRPM5 to be the most downregulated gene in WFS1 mice. While this gene is involved in thermal reception, it is also involved in taste perception, and it is also involved in diabetes and insulin secretion [72]. Our RNA seq study combined the changes in the TRPM5 expression with the reduced insulin secretion in *WFS1*-deficient pancreatic cells [67]. And, again, many of the differentially expressed genes in pancreatic islets were related to the proteasomal and lysosomal pathways, suggesting the common pathogenic mechanism for the cellular damage.

In the most recent transcriptomic study using WFS1 mouse retinas, significant changes in GFAP, C4A, and C4B were described [57]. Interestingly, this is a very wide overlap with the results from the gene chip-based analysis of the hypothalamus, where we detected the same three genes changed in *WFS1* mutant mice [59]. This result indicates again that independent studies performed by different groups but using identical models with identical molecular definition with the same mutation, could help to develop robust functional genomic understanding of the genomic pathogenesis of the diseases.

Transcriptomic studies with different *WFS1* mutant mouse tissues have been challenging, as they have been conducted with different technologies and with different levels of the available annotation. Gene chips have come a long way from the years when we performed the studies with WFS1 mouse tissues. Annotation quality is much better now, and our background knowledge of the genome and different genes has improved. Similarly, RNA sequencing technology provides a much better dynamic range and enables detecting the genes with variable expression levels. These have been main challenges we have faced during our studies. Fortunately, stringent formal statistical analysis, combined with functional annotation, helps to overcome the drawbacks of different genomic technologies.

In addition to the transcriptomic analysis, we and others have performed proteomic studies using the WFS1 mouse hippocampus, retinas, and muscles [14,57,73,74]. In the hippocampus, pro-SAAS peptide fragments were highly and significantly upregulated in *WFS1* mutant mice, indicating the alteration in the peptide processing [73]. The proteomic analysis of the muscle tissue identified significant alterations related to energy metabolism and oxidative phosphorylation. Proteomic analysis of retinas identified many similar proteins differentially expressed, as with RNAseq, indicating an excellent overlap between different technologies and confirming the RNA-seq results [57]. These findings coincided with the physiological and biochemical experimental data [14,73,74].

## 7. Evidence for the Neurodegeneration

While all earlier studies already suggested the direct impact of the *WFS1* gene on neurodegeneration, only recent studies have generated strong experimental evidence to support the direct role of the *WFS1* gene in neurodegeneration. In a very comprehensive biochemical study, the interaction of WFS1 with tau aggregation was identified [63]. This finding helps to explain the profound neurodegeneration seen in WFS1 patients and, most importantly, again indicates the degenerative mechanism of WFS1 to be independent of the mitochondrional dysfunction [58]. Previous publications also support the *WFS1* involvement in retinal and optic nerve degeneration and brain atrophy, but this paper was direct evidence for the WFS1 involvement in the aggregation of other proteins. This means that the reduced or deleted function of *WFS1* that causes increased ER stress causes the aggregation of other proteins inside the cells and induces cell death by the ER stress and unfolded protein mechanism. Moreover, administration of *WFS1* delayed or even completely blocked the progression of neurodegeneration [63]. This is a significant finding and shows the quantitative relationship between the functional decline of the *WFS1* gene and neurodegeneration. Therefore, we can conclude that *WFS1* deficiency is directly related to the neurodegeneration, and we have also shown the potential of the *WFS1* gene as a therapeutic opportunity for the neurodegeneration and dementia.

## 8. Discussion

WFS1 syndrome is a rare disorder, but the studies on this disease have revealed pathogenetic mechanisms that are relevant for other diseases and common conditions. WFS1 syndrome and its impact on type 2 diabetes research is well established, and many GWAS studies have shown the impact of *WFS1* variations in the human population in the polygenic risk of type 2 diabetes [75,76,77]. *WFS1* is also related to type 1 diabetes, hearing loss, blindness, and different eye problems. All these syndromes can be deduced from the original clinical features of WFS1. The neurodegeneration, on the other hand, seems obvious, but has been a difficult task to tackle. The clinical data of WFS1 supported the existence of brain atrophy and neurodegeneration [11,16,78,79,80]. However, the problem with such a complex phenotype is that it is hard to focus on a particular trait or even to see the common mechanism connecting all these symptoms and diseases.

A few events have been instrumental to advance WFS1 research significantly. First, the development of a straightforward and robust mouse model providing a reproducible tool for genomic and molecular studies. Second, development of the genomic technologies enabled analysis of multiple tissue samples within a short time and combining it with reproducible bioinformatic solutions. This has enabled obtaining comprehensive description of the molecular changes underlying WFS1. Finally, many groups continued to work on the drugs for the syndrome, and different solutions have been published. CRISPR-Cas9 correction of the mutation of the patient’s IPSC-derived β cells rescued streptozocin-induced diabetes in mice and restored insulin secretion, showing the proof of the concept [81]. Another good example is that using of the calpain inhibitor, ibudilast, reversed the effect of WFS1 mutation and normalised the function of rat insulinoma cells [82]. Therefore, genomic information, combined with detailed biochemical and molecular experiments, has paved a way toward the potential therapeutic opportunities for WFS1. And, most recently, sigma 1 receptor activation with PRE-084 alleviated the symptoms of WFS1 in preclinical models [58].

In addition to these experimental opportunities, several ongoing clinical trials evidence the continued interest to translate these findings as soon as possible. Valproate, dantrolene, tirzepatide, and AMX0035 are drugs currently in clinical trials offering the experimental treatment possibility for patients with WFS1 syndrome. These drugs are already clinically approved for more common conditions, and these trials are mostly repurposing trials to test the efficacy of the existing treatments. Repurposing was possible thanks to the existing experimental data from the molecular and genomic studies.

Gene therapy is an emerging technology for offering a solution for WFS1. This can include either transfer of the *WFS1* gene itself or cell survival factors, like MANF [83,84]. Both strategies have been experimentally proved, and the principle is viable. Considering the recent success with gene transfer technologies and improved delivery systems, the solution for the WFS1 could be RNA-based therapeutics. This can be achieved by viral delivery systems, like AAV, or even with the nanoparticle-based RNA delivery options. Nanoparticles have proven to be very useful and successful technologies. On the other hand, *WFS1* gene transfer would be suitable not only for WFS1 itself, but also for other more common neurodegenerative, deafness, or diabetic syndromes. *WFS1* gene transfer has been shown to be efficient against tau aggregation and can, therefore, stop the progression of dementia [63]. Many research groups are working on specific technological solutions to restore the function of the *WFS1* gene or support the cellular resilience.

Genomic studies have proven to be efficient and comprehensive for understanding the pathogenesis of the diseases and form fundamentals for genomic pathology and precision therapies. WFS1 is a good example of how the studies on one rare disease generate more general knowledge that is applicable and translatable for more common conditions, like neurodegenerative diseases and diabetes in the case of WFS1. However, the impact of *WFS1* mutation on muscular dystrophies remains to be explored. Patients with WFS1 may have muscular dystrophies and cardiomyopathy, and, therefore, the muscular phenotype requires more attention [13,85]. Supportive therapy for the muscular system might be necessary to improve the quality of life of patients.

## 9. Conclusions

Studies with WFS1 have extended our understanding of the pathogenesis of the syndrome and other similar syndromes with ER stress and UPR response activation. ER stress has become a universal pathological mechanism for the degenerative diseases, and it can trigger multiple reactive responses, like inflammation and autophagy. The genomic studies on WFS1 have unveiled the molecular mechanisms and pathways that are involved in this type of pathology. While the phenotype of the syndrome is complex, the use of animal models, different tissue sets, and variable analytical tools can give a comprehensive description of the genetic networks that have been altered by the mutation.

## 10. Future Directions

WFS1 is a devastating syndrome caused by mutations in a pleiotropic gene. The variability of the clinical syndrome and the rarity of the condition have made the research efforts hard to advance the field. However, the mutant mouse models have helped to obtain better understanding of the syndrome, and the research has led to the comprehensive description of the molecular pathways affected by this syndrome. Thanks to these efforts, we are closer than before to the effective treatment of WFS1. Moreover, it appears that effective treatment for WFS1 could also be repurposed for common conditions, like type 2 diabetes and neurodegenerative conditions. Different experimental solutions have been proposed and developed to treat WFS1 syndrome or WFS1-related syndromes. Intensive translational programmes are necessary to push these experimental solutions over the line of clinical practice.

## Figures and Tables

**Figure 1 biomolecules-13-01346-f001:**
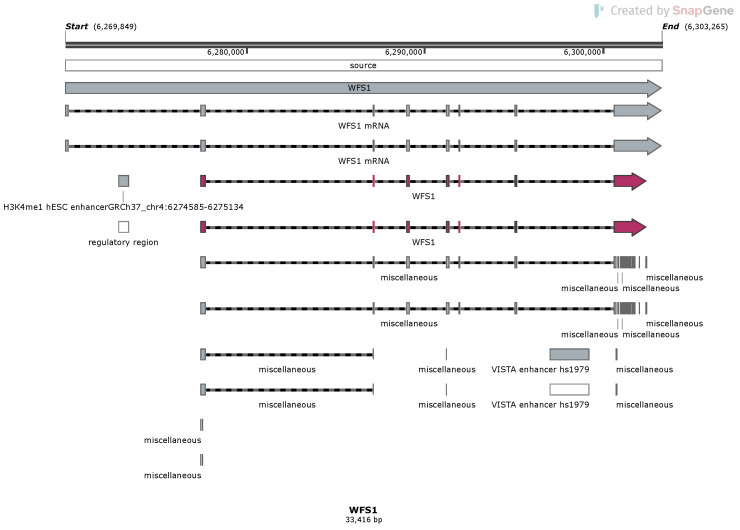
Overview of the structure of the *WFS1* gene and its two different transcripts with genomic signals. Red boxes and arrows indicate two CDS regions. Different genomic signals (enhancers, regulatory region) are indicated by boxes.

**Figure 2 biomolecules-13-01346-f002:**
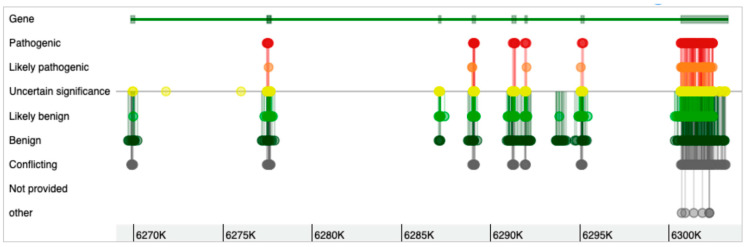
Genetic variants in the *WFS1* gene and their clinical significance. Exon 8 has most of the pathogenic variants of the *WFS1* gene.

**Table 1 biomolecules-13-01346-t001:** Different transcripts of the *WFS1* gene and respective proteins.

Transcript ID	Name	bp	Protein	Biotype	UniProt Match
ENST00000226760.5	WFS1-201	3640	890aa	Protein coding	A0A0S2Z4V6 O76024
ENST00000682275.1	WFS1-211	3662	902aa	Protein coding	A0A669KAX3
ENST00000673991.1	WFS1-208	3633	902aa	Protein coding	A0A669KAX3
ENST00000684087.1	WFS1-214	3602	890aa	Protein coding	A0A0S2Z4V6 O76024
ENST00000683395.1	WFS1-212	3446	278aa	Protein coding	A0A804HIL2
ENST00000503569.5	WFS1-202	3255	890aa	Protein coding	A0A0S2Z4V6 O76024
ENST00000506362.2	WFS1-203	2882	807aa	Protein coding	H0Y9G5
ENST00000673642.1	WFS1-207	2013	427aa	Protein coding	A0A669KBF0
ENST00000684700.1	WFS1-215	917	234aa	Protein coding	A0A804HK77
ENST00000682059.1	WFS1-210	718	164aa	Protein coding	A0A804HKM5
ENST00000684054.1	WFS1-213	553	157aa	Protein coding	A0A804HIL0
ENST00000674051.1	WFS1-209	451	148aa	Protein coding	A0A669KB26
ENST00000507765.1	WFS1-205	3655	No protein	Retained intron	-
ENST00000506588.6	WFS1-204	1439	No protein	Retained intron	-
ENST00000513395.1	WFS1-206	570	No protein	Retained intron	-

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
