# Peer review of "Genomics of Wolfram Syndrome 1 (WFS1)"

_biomolecules, 2023, doi:10.3390/biom13091346_

Round 1
Reviewer 1 Report
The manuscript is very interesting and well written. It offers an extensive review of the pathogenic mechanisms of Wolfram Syndrome 1.
Author Response
Thank you very much for your comments, very much appreciated.
Reviewer 2 Report
This review about Wolfram syndrome 1 is definitely well-written. The author has organized the manuscript in paragraphs, using coloured-figures as clear as understandable and well-structured tables. This review makes the topic about WFS1 genetics almost all-embracing, including mutations, animal models, current studies. References are updated. Discussion and conclusion are not redundant but simple enough to convey the message of the author.
Author Response
Thank you very much for your positive comments, very much appreciated.
Reviewer 3 Report
It is a interesting study. Syndrome is very rare and unknown. A comprehensive review with more data. It is a good ideas to insist on article to gene terapy of this syndrome. The figure is not so clear.
Need to improve. Figure, discussion, more example of gene therapy
Author Response
Thank you very much for your positive comments.
I have added more information to the figure legends and replaced Figure 1 with a new version after removing the restriction enzyme sites.
I also added a paragraph about the gene therapy opportunities and added appropriate references.
Hopefully, I was able to improve the manuscript and this version is now acceptable.
Round 2
Reviewer 3 Report
The articol is revised. I dont have other requerements
Kind regards